# Aberrant hippocampal Ca²⁺ microwaves following synapsin-dependent adeno-associated viral expression of Ca²⁺ indicators

**Nicola Masala[1,2,3†], Manuel Mittag[4], Eleonora Ambrad Giovannetti[4], Darik A O'Neil[5], Fabian J Distler[1,2], Peter Rupprecht[6,7], Fritjof Helmchen[6,7], Rafael Yuste[5], Martin Fuhrmann[4], Heinz Beck[1,2,8], Michael Wenzel[1,2,3]\*, Tony Kelly[1,2]\***

[1]University of Bonn, Faculty of Medicine, Institute for Experimental Epileptology and Cognition Research (IEECR), Bonn, Germany; [2]University Hospital Bonn, Bonn, Germany; [3]Department of Epileptology, University Hospital Bonn, Bonn, Germany; [4]Neuroimmunology and Imaging Group, German Center for Neurodegenerative Diseases (DZNE), Bonn, Germany; [5]NeuroTechnology Center, Columbia University, New York, United States; [6]Brain Research Institute, University of Zurich, Zurich, Switzerland; [7]Neuroscience Center Zurich, University of Zurich, Zurich, Switzerland; [8]German Center for Neurodegenerative Diseases (DZNE), Bonn, Germany

**\*For correspondence:**
michael.wenzel@ukbonn.de (MW);
tkelly@uni-bonn.de (TK)

**Present address:** †Department of Anatomy and Neurobiology, University of California, Irvine, Irvine, United States

**Competing interest:** The authors declare that no competing interests exist.

**Abstract** Genetically encoded calcium indicators (GECIs) such as GCaMP are invaluable tools in neuroscience to monitor neuronal activity using optical imaging. The viral transduction of GECIs is commonly used to target expression to specific brain regions, can be conveniently used with any mouse strain of interest without the need for prior crossing with a GECI mouse line, and avoids potential hazards due to the chronic expression of GECIs during development. A key requirement for monitoring neuronal activity with an indicator is that the indicator itself minimally affects activity. Here, using common adeno-associated viral (AAV) transduction procedures, we describe spatially confined aberrant Ca²⁺ microwaves slowly travelling through the hippocampus following expression of GCaMP6, GCaMP7, or R-CaMP1.07 driven by the synapsin promoter with AAV-dependent gene transfer in a titre-dependent fashion. Ca²⁺ microwaves developed in hippocampal CA1 and CA3, but not dentate gyrus nor neocortex, were typically first observed at 4 wk after viral transduction, and persisted up to at least 8 wk. The phenomenon was robust and observed across laboratories with various experimenters and setups. Our results indicate that aberrant hippocampal Ca²⁺ microwaves depend on the promoter and viral titre of the GECI, density of expression, as well as the targeted brain region. We used an alternative viral transduction method of GCaMP which avoids this artefact. The results show that commonly used Ca²⁺-indicator AAV transduction procedures can produce artefactual Ca²⁺ responses. Our aim is to raise awareness in the field of these artefactual transduction-induced Ca²⁺ microwaves, and we provide a potential solution.

## eLife assessment

This **important** study provides **convincing** evidence of artefactual calcium microwaves during calcium imaging of populations of neurons in the hippocampus using methods that are common in the field. The work raises awareness of these artefacts so that any research labs planning to do calcium imaging in the hippocampus can avoid them by using alternative strategies that the authors propose.

## Introduction

There has been an explosion in the use of imaging techniques to record neuronal activity over the past 30 y, starting with the introduction of organic calcium indicators to measure neuronal population activity (*Yuste and Katz, 1991*) and accelerated by rapid advances in the development of genetically encoded $Ca^{2+}$ indicators (GECIs) (*Miyawaki et al., 1997*). Specific advantages of neuronal $Ca^{2+}$ imaging with GECIs lie in the ability of chronic cellular scale recordings of sizeable, densely labelled neuronal or glial populations with subtype specificity, without having to perturb the cell membrane or add a synthetic chemical to the brain (*Grienberger and Konnerth, 2012*; *Rose et al., 2014*; *Semyanov et al., 2020*).

Commonly used GECIs such as the GCaMP family have been continually improved since their initial development (*Nakai et al., 2001*), offering high signal-to-noise ratio, sensitivity, and response kinetics such that they can detect single-action potentials in vivo. This allows the reporting of cellular activity as well as the activity of sub-compartments such as the dendritic arbour (*Chen et al., 2013*; *Dana et al., 2019*; *Zhang et al., 2023*). Typically, GCaMP is expressed using transgenic animals or adeno-associated viral (AAV) transduction techniques (*Tian et al., 2012*; also see *Grødem et al., 2023*). The use of transgenic animals has the advantage of not requiring AAV transduction, thus reducing surgery load for animals and likelihood of indicator overexpression. In contrast, AAV GECI transduction is straightforward (breeding/crossing not required), can be targeted to virtually any brain region, and typically offers enhanced fluorescence (due to higher expression levels). Furthermore, AAV transduction avoids potential hazards due to chronic GECI expression during development.

While offering unprecedented new insights into cellular-scale neuronal network dynamics, it has also been reported that GECI expression in neurons can result in unwanted side effects. Depending on the expression approach, neurons have shown reduced dendritic branching and impairment in cell health, leading to cytotoxicity and cell death (*Gasterstädt et al., 2020*; *Resendez et al., 2016*). Furthermore, increased $Ca^{2+}$ buffering due to the addition of $Ca^{2+}$ indicators has been associated with alterations in intracellular $Ca^{2+}$ dynamics (*Grienberger and Konnerth, 2012*; *McMahon and Jackson, 2018*). In addition, chronic expression of GCaMP can lead to accumulation in the nucleus and changes in gene expression (*Yang et al., 2018*). Again, depending on the specific expression approach, GCaMP variant, and experimental time course, such changes may alter cellular physiology and excitability. For example, increased firing rates have been observed in hippocampal neurons expressing GCaMP5G from CaMKIIa-Cre; PC::G5-tdT mice, and epileptiform activity in neocortex in some GCaMP6-expressing transgenic mice (*Gee et al., 2014*; *Steinmetz et al., 2017*).

Here, we describe microscale $Ca^{2+}$ waves that are highly confined in space and progress slowly through the hippocampus following local GCaMP or R-CaMP viral transduction. Such aberrant hippocampal waves were typically first observed 4 wk following injection of commercially available AAVs expressing GCaMP6, GCaMP7, or R-CaMP1.07 under the synapsin promoter. The phenomenon occurred upon GECI transduction in CA1 and CA3, but not in dentate gyrus (DG) nor neocortex, was robustly observed across laboratories with various experimenters and setups, and highlights the necessity of careful use of transduction methods and control measures. Reducing the transduction titre diminished the likelihood of aberrant hippocampal $Ca^{2+}$ waves, and an alternative viral transduction method employing sparser and Cre-dependent GCaMP6s expression in principal cells avoided the aberrant $Ca^{2+}$ waves. Furthermore, in three transgenic GCaMP mouse lines (thy1-GCaMP6s or 6f; Vglut1-IRES2-Cre-D × Ai162(TIT2L-GC6s-ICL-tTA2)), aberrant $Ca^{2+}$ microwaves were never observed. The aim of this article is to raise awareness in the field of artefactual transduction-induced $Ca^{2+}$ waves and encourage others to carefully evaluate their $Ca^{2+}$ indicator expression approach before embarking on chronic in vivo calcium imaging of the hippocampus.

## Results

### Aberrant $Ca^{2+}$ microwave progression through the hippocampus

Based on published protocols, we injected AAV1 particles (pAAV.Syn.GCaMP6s.WPRE.SV40, Addgene #100843, titre $1 \times 10^{13}$ vg/ml) into the hippocampus (total injection volume: 500 nl undiluted [1:1] virus solution) of C57BL/6 wildtype animals (6 weeks old) and performed in vivo two-photon imaging to record cellular activity at 2, 4, and 6–8 wk post-injection (p.i.) (*Figure 1a*). Viral transduction resulted in GCaMP6s expression throughout the hippocampal CA1, CA3, and DG areas under

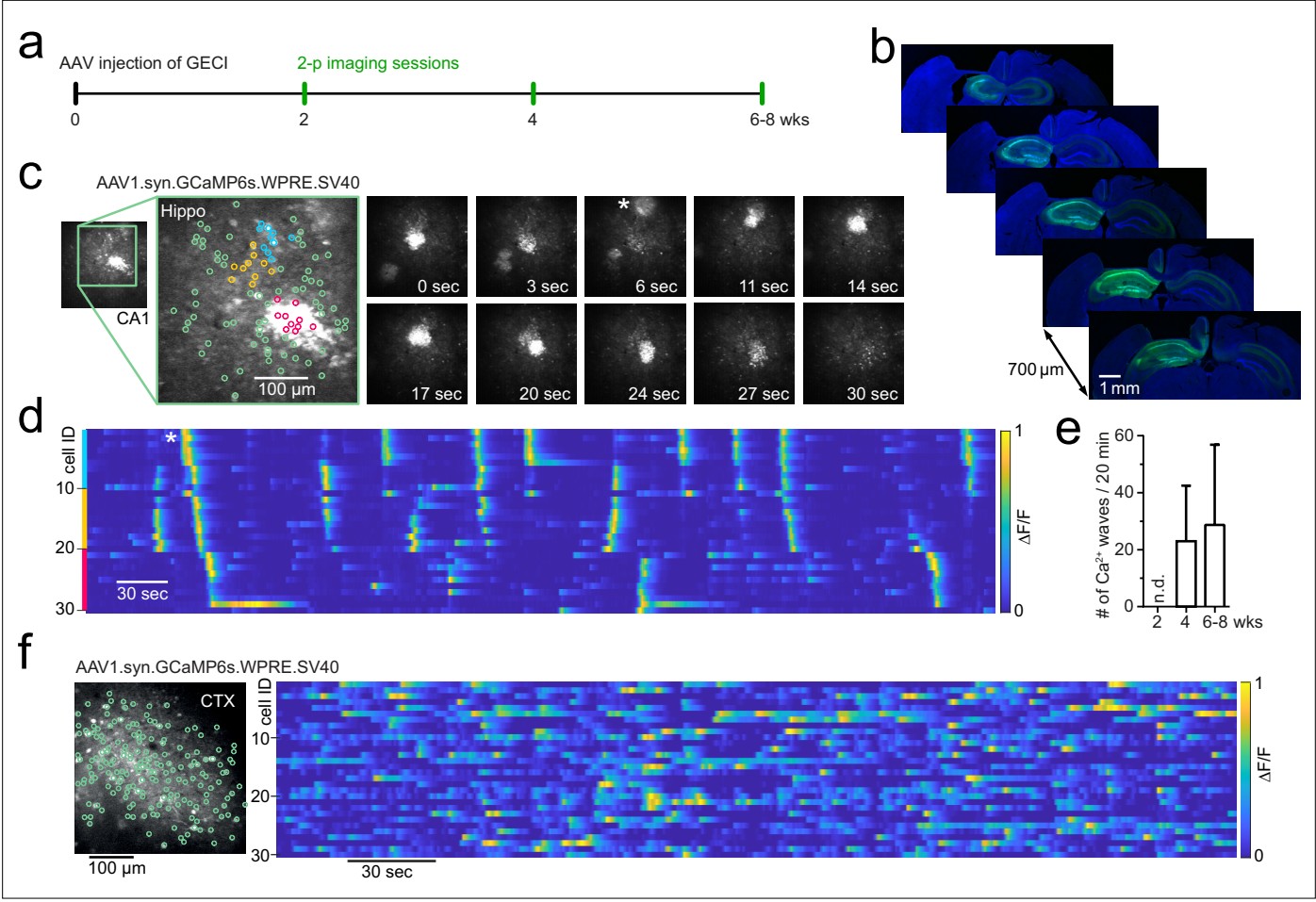

**Figure 1.** Development of Ca²⁺ microwaves travelling through hippocampus following GCaMP transduction. (**a**) Experimental protocol to examine CA1 neuronal activity using two-photon imaging following adeno-associated viral (AAV) transduction of genetically encoded Ca²⁺ indicators. (**b**) Immunohistochemical sections following the last imaging session. GCaMP6s (AAV1.syn.GCaMP6s.SV40, Addgene #100843) expression throughout the ipsilateral hippocampus and projection pathways in the contralateral hippocampus. (**c**) Two-photon Ca²⁺ imaging of field of view (FOV) in CA1 at 4 wk post-injection (p.i.) showing aberrant Ca²⁺ microwaves (see also *Video 1*). Magnified inset shows three coloured neuronal subgroups (blue, orange, magenta) based on their spatial vicinity from a total population of 100 identified neurons (green). *Right*: time series of two-photon Ca²⁺ imaging FOVs showing two Ca²⁺ microwaves, the first at 0 s, the second appearing at 6 s (asterisk). The second wave progresses through FOV over dozens of seconds. (**d**) Raster plot of individual neuronal Ca²⁺ activity (ΔF/F, 1 min moving window, traces max-normalized per neuron) from neighbouring subgroups (colours correspond to **c**). Asterisk (same as in **c**): a Ca²⁺ microwave advances through neighbouring neuronal subgroups. (**e**) Occurrence rate (mean ± 95% CI) of aberrant Ca²⁺ microwaves with increasing expression time, following viral transduction of AAV1.syn.GCaMP6s.SV40 in mature C57BL/6 wildtype animals (n=4). n.d. = none detected. (**f**) Two-photon Ca²⁺ imaging FOV in the visual cortex at 6 wk p.i. (left) with normal sparse spontaneous Ca²⁺ activity and no detected Ca²⁺ microwaves (right; raster plot of ΔF/F, 1 min moving window, traces max-normalized per neuron).

the imaging window (*Figure 1b*). As expected, the expression was primarily restricted to the ipsilateral hippocampus, with some labelling of projection pathways also in the contralateral hippocampus. There was no evidence of gross transduction-related morphological changes to the hippocampus (see *Figure 1b*), with no changes in CA1 pyramidal cell layer thickness or CA1 thickness (pyramidal layer thickness: 49 ± 12.5 μm ipsilateral and 50.3 ± 11.1 μm contralateral, n = 4, Student's *t*-test p=0.89; CA1 thickness: 553.3 ± 14 μm ipsilateral and 555.8 ± 62 μm contralateral, n = 4, Student's *t*-test p=0.94; 48 ± 13 wk p.i. at the time of perfusion). At 4 wk after injection, a time point commonly used for imaging cellular activity, we observed distinctive aberrant microscale Ca²⁺ waves that travelled through CA1 recruiting neighbouring cells (*Figure 1c and d*, *Video 1*, n = 4 mice). Ca²⁺ microwaves were maintained up to 6–8 wk after AAV injections (*Figure 1e*, n = 4 mice). In wildtype mice, these Ca²⁺ microwaves were not observed at an earlier time point (2 wk p.i.; p<0.05 using Kruskal–Wallis *H*-test for comparison between the three time points).

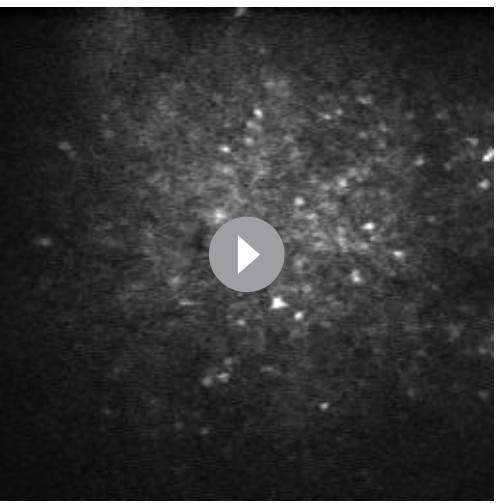

**Video 1.** GCaMP6s two-photon calcium imaging in the hippocampal CA1 region, around 100 µm beneath the hippocampal surface (stratum pyramidale), FOV ~600x600 µm, ~4 wk after transduction of AAV1 particles containing pAAV.Syn.GCaMP6s.WPRE.SV40 (Addgene plasmid #100843) in a mature bl6 wildtype mouse. Imaging wavelength = 940 nm, acquisition speed = 15 frames/s. Movie played at ×5 acquisition speed. Imaging was performed at the IEECR/University of Bonn.

https://elifesciences.org/articles/93804/figures#video1

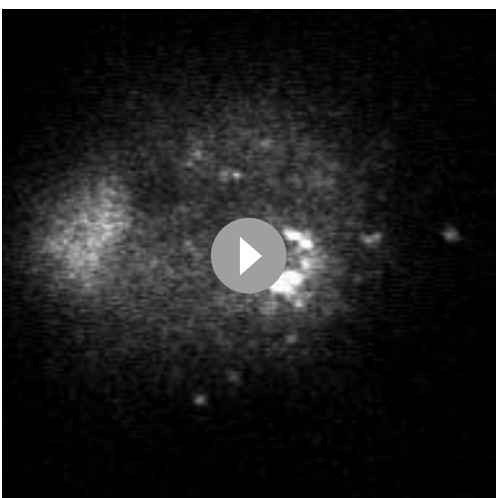

**Video 2.** GCaMP6s two-photon calcium imaging in the hippocampal CA1 region, around 100 µm beneath the hippocampal surface (stratum pyramidale), FOV ~450x450 µm, ~7 wk after transduction of AAV1 particles containing pAAV.Syn.GCaMP6s.WPRE. SV40 (Addgene plasmid #100843) in an ~3-month-old transgenic mouse (same as in *Video 2*; *Scn2a*[A263V] model of genetic epilepsy). Imaging wave length = 940 nm, acquisition speed = 15 frames/s. Movie played at ×5 acquisition speed. Imaging was performed at the IEECR/University of Bonn.

https://elifesciences.org/articles/93804/figures#video2

The properties of the $Ca^{2+}$ microwaves depended on the hippocampal region and exact recording location. For instance, although the $Ca^{2+}$ waves were consistently observed in CA1, the spatial dimensions of the $Ca^{2+}$ microwaves were broader in the stratum oriens compared with stratum pyramidale of CA1 (*Videos 2* [str. pyr.] and *Video 3* [str. oriens]), which likely reflects concomitant neuropil activation. We next examined whether the CA1 network is particularly prone to the generation of such waves and whether they show regional specificity. Upon viral GCaMP6s transduction under synapsin, $Ca^{2+}$ waves were observed in both CA1 (n = 4/4; *Videos 1–5*) and CA3 (n = 1/1; *Video 6*), but interestingly, not in the DG (n = 3 mice, 4, 8, and 10 wk p.i., 40 min total recording time per mouse). In contrast to hippocampus, synapsin-dependent GCaMP6s expression restricted to the neocortex (V1 or somatosensory cortices) did not result in cortical $Ca^{2+}$ waves in our hands (*Figure 1f*, n > 20 mice).

## Aberrant $Ca^{2+}$ microwaves in disease models

The observed $Ca^{2+}$ microwaves were distinct from local seizure activity (no rhythmicity, no typical ictal evolution, no postictal depression) (*Masala et al., 2023*; *Muldoon et al., 2015*; *Wenzel et al., 2017*; *Wenzel et al., 2019a*) and spreading depolarization/depression phenomena (no concentric expansion, no post-wave neural depression). However, the occurrence of these artificial events may be confused as aberrant activity related to a pathology, especially when studying pathologies with known cellular and network hyperexcitability. For example, we initially found the aberrant hippocampal $Ca^{2+}$ microwaves in the *Scn2a*[A263V] model of genetic epilepsy; however, these $Ca^{2+}$ waves in CA1 of heterozygous animals (5/5 mice) were in general similar to those detected in wild-type animals at 4 wk p.i. In the *Scn2a*[A263V] model, in one case (1/5 animals), $Ca^{2+}$ waves were observed even at 2 wk p.i. (*Video 4*). Furthermore, hippocampal transduction of jGCaMP7f under synapsin (Addgene #104488, AAV9 particles, original titre 2.5 × 10^13 vg/ml, total injection volume 1000 nl [1:2 dilution]) in a mouse model of Alzheimer's disease (PV-Cre::APPswe/PS1dE9) also resulted in $Ca^{2+}$ microwaves (n = 3/6 mice). Together, these experiments show that common AAV injection procedures of GECIs under the synapsin promoter lead to artefactual hippocampal $Ca^{2+}$ microwaves in wildtype mice and genetic mouse models of disease.

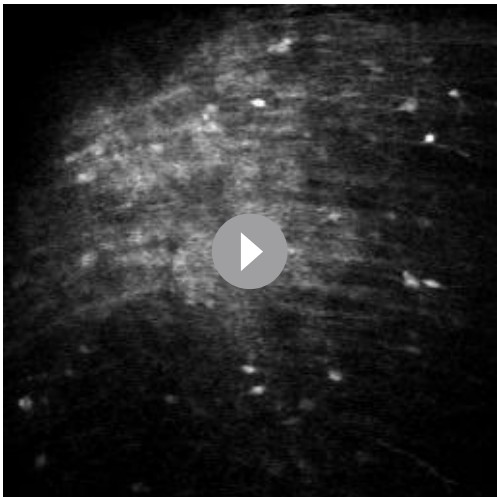

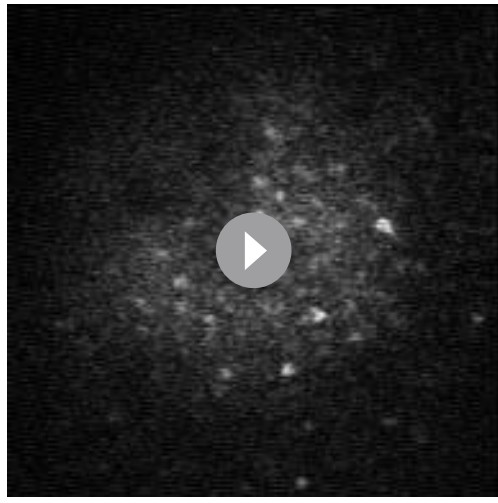

**Video 3.** Same animal (*Scn2a*^A263V model of genetic epilepsy) and time point of imaging as in *Video 4*. GCaMP6s two-photon calcium imaging in the hippocampal CA1 region, around 25 μm beneath the hippocampal surface (stratum oriens), FOV ~350x350 μm, ~7 wk after transduction of AAV1 particles containing pAAV.Syn.GCaMP6s.WPRE.SV40 (Addgene plasmid #100843). Imaging wavelength = 940 nm, acquisition speed = 15 frames/s. Movie played at ×5 acquisition speed. Imaging was performed at the IEECR/University of Bonn.

https://elifesciences.org/articles/93804/figures#video3

**Video 4.** GCaMP6s two-photon calcium imaging in the hippocampal CA1 region, around 100 μm beneath the hippocampal surface (stratum pyramidale), FOV ~450x450 μm, ~2 wk after transduction of AAV1 particles containing pAAV.Syn.GCaMP6s.WPRE. SV40 (Addgene plasmid #100843) in an ~2-month-old transgenic mouse model of genetic epilepsy (heterozygous *Scn2a*^A263V mouse). Imaging wavelength = 940 nm, acquisition speed = 15 frames/s. Movie played at ×5 acquisition speed. Imaging was performed at the IEECR/University of Bonn.

https://elifesciences.org/articles/93804/figures#video4

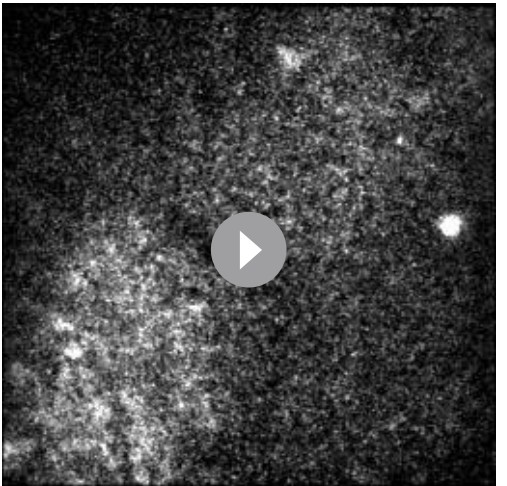

**Video 5.** R-CaMP1.07 two-photon calcium imaging in the hippocampal CA1 region, around 100 μm beneath the hippocampal surface (stratum pyramidale), FOV ~200x200 μm, ~10 wk after transduction of AAV1 particles containing ssAAV-9/2-hSyn1-chI-RCaMP1.07-WPRE-SV40p(A) (Viral Vector Core UZH #V224-9) in a mature (~5 mo) bl6 wildtype mouse. Imaging wavelength = 960 nm, acquisition speed = 30.88 frames/s. Movie played at ×5 acquisition speed. Imaging was performed at the Neuroscience Center Zurich (UZH).

https://elifesciences.org/articles/93804/figures#video5

## Properties and robustness of aberrant hippocampal Ca²⁺ waves

Next, we investigated the robustness of the aberrant $Ca^{2+}$ microwaves across institutes and conditions. We chose to compare the incidence of aberrant $Ca^{2+}$ microwaves in the CA1 region in four separate institutes in three different countries following transduction of GCaMP6s (Addgene #100843; IEECR/UoB, CU), GCaMP6m or jGCaMP7f (Addgene #100841 or #104488; DZNE), or RCaMP1.07 (Viral Vector Facility UZH #V224-9; UZH, *Video 5*; *Table 1*).

The incidence of aberrant hippocampal $Ca^{2+}$ microwaves was robust, observed at the four different institutes each using variations of commonly used, published viral transduction procedures and standard two-photon $Ca^{2+}$ imaging protocols (*Table 1*; see 'Materials and methods' for more details). Importantly, aside from the targeted region, the viral titre was important as halving the original AAV1.syn. GCaMP6m viral titre decreased the number of animals that developed aberrant $Ca^{2+}$ microwaves from 80% of animals (4/5, original titre, $1 \times 10^{13}$ vg/ml) to 43% of animals (3/7, 50% reduced titre, $0.5 \times 10^{13}$ vg/ml) (see *Table 1*). To statistically test

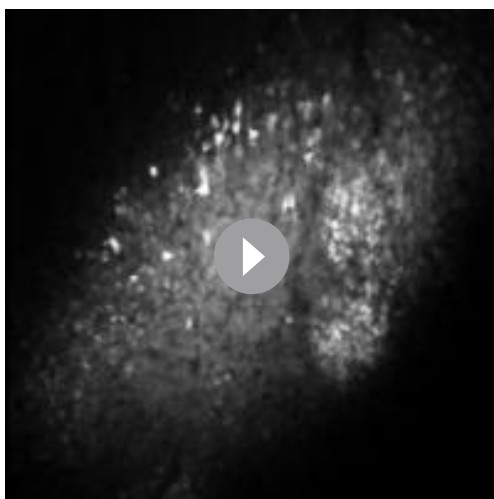

**Video 6.** GCaMP6s two-photon calcium imaging in the hippocampal CA3 region, stratum pyramidale, FOV ~600x600 μm, ~7 wk after transduction of AAV1 particles containing pAAV.Syn.GCaMP6s.WPRE.SV40 (Addgene plasmid #100843) in a mature bl6 wildtype mouse. Imaging wavelength = 940 nm, acquisition speed = 30.206 frames/s. Movie played at ×5 acquisition speed. Imaging was performed at Columbia University.

https://elifesciences.org/articles/93804/figures#video6

the involvement of expression level, we used a generalized linear model. For injections into CA1 in the hippocampus (n = 28), a multivariate logistic GLM (Ca$^{2+}$ wave ~ dilution + p.i. wk) found both dilution and p.i. weeks were significantly related to Ca$^{2+}$ wave incidence (model deviation above null = 7.5; dilution: z score = 2.18, p<0.05; p.i. wk: z score = 2.22, p<0.05).

To examine how robust the Ca$^{2+}$ microwaves properties were, we compared properties across the different laboratories following expression of GCaMP6m and GCaMP6s variants. The occurrence rate of the aberrant Ca$^{2+}$ microwaves was similar across the different institutes (*Figure 2a*). Ca$^{2+}$ microwaves were spatially confined (diameter range of 200–300 μm), moved across the field of view (FOV) with slow progression speeds (speed range of 5–25 μm/s) (*Figure 2b*), and displayed no rhythmicity but rather plateau-like Ca$^{2+}$ activity. No statistical differences were observed in Ca$^{2+}$ microwave properties between the different institutes, suggesting that these values provide a reasonable range. In addition, the Ca$^{2+}$ microwaves were not restricted to a single GECI variant or version, with Ca$^{2+}$ waves observed following expression of GCaMP6m (n = 4), GCaMP6s (n = 5), and GCaMP7f (n = 3), as well as R-CaMP1.07 (n = 1) (*Figure 2c* and *Table 1*).

In summary, upon synapsin-promoter-dependent AAV Ca$^{2+}$ indicator expression, depending on the time of expression and viral transduction titre, Ca$^{2+}$ microwaves were specifically observed in the CA1 and CA3 subregions of the hippocampus. For CA1, the Ca$^{2+}$ microwaves were observed across laboratories and countries and animal models using common transduction procedures (for an overview, see *Table 1*).

## Alternative transduction method of GCaMP to avoid aberrant Ca$^{2+}$ microwaves

In light of these results, we tested an alternative expression approach to avoid aberrant hippocampal Ca$^{2+}$ microwaves. To this end, we selected an approach to both limit the expression to principal cells and label a sparse population of the principal cells using a dual AAV injection approach. Here, Cre-dependent expression of GCaMP6s or GCaMP6m was achieved in a sparse population of principal cells under the CaMKII promoter (AAV1.syn.Flex.GCaMP6s.WPRE.SV40, Addgene #100845, and AAV1.CamKII0.4.Cre.SV40, Addgene #105558; n = 2 or, AAV1.Syn.Flex.GCaMP6m.WPRE.SV40, Addgene #100838 and AAV9.CamKII0.4.Cre.SV40, Addgene #105558; n = 3; *Figure 2d*; see *O'Hare et al., 2022*; *Jimenez et al., 2020*; *Sheffield and Dombeck, 2015*), upon which no Ca$^{2+}$ microwaves were observed (0/5 animals, *Figure 2d*). Furthermore, hippocampal Ca$^{2+}$ microwaves were neither observed in transgenic thy1-GCaMP6s nor 6f mice (JAX strain 025776 or 024276; up to 3 mo of chronic imaging in n > 30 mature mice age > p60, cumulative imaging time >200 hr), nor in Vglut1-IRES2-Cre-D mice crossed with Ai162(TIT2L-GC6s-ICL-tTA2)-D mice (JAX strains 037512, 031562; up to 3 mo of chronic imaging in n = 5 mature mice > p60).

## Discussion

Here we report titre- and expression-time-dependent aberrant hippocampal Ca$^{2+}$ microwaves in CA1 and CA3 regions following viral expression of GCaMP or R-CaMP1.07 under the synapsin promoter.

**Table 1.** Viruses used for the expression of genetically encoded calcium indicators (GECIs).

Viral titre is from Addgene documentation and was used at original concentration (dilution of 1:1) or at a dilution of 1:2. Syn.Flex. GCaMP6s and CamKII0.4.Cre were co-injected and therefore diluted to 1:2. Two-photon $Ca^{2+}$ imaging was performed from 2 wk after injection in the hippocampus (CA1, CA3, or DG) or neocortex (Ctx). $Ca^{2+}$ microwave incidence was determined from the number of animals exhibiting $Ca^{2+}$ microwaves at the specified time point and region.

| AAV | Construct | Source (Addgene id) | Original titre (vg/ml) | Dilution | Injection volume | Post-injection (wk) | Region | $Ca^{2+}$-wave incidence (%) | n | Mouse model | Institute |
|---|---|---|---|---|---|---|---|---|---|---|---|
| AAV1 | Syn.GCaMP6s | 100843 | $1 \times 10^{13}$ | 1:1 | 0.5 | 2 | CA1 | 0 | 0/4 | wt | UoB |
| AAV1 | Syn.GCaMP6s | 100843 | $1 \times 10^{13}$ | 1:1 | 0.5 | 4–6 | CA1 | 100 | 4/4 | wt | UoB |
| AAV1 | Syn.GCaMP6s | 100843 | $1 \times 10^{13}$ | 1:1 | 0.5 | 2 | CA1 | 20 | 1/5 | Scn2a* | UoB |
| AAV1 | Syn.GCaMP6s | 100843 | $1 \times 10^{13}$ | 1:1 | 0.5 | 4–8 | CA1 | 100 | 5/5 | Scn2a* | UoB |
| AAV1 | Syn.GCaMP6m | 100841 | $1 \times 10^{13}$ | 1:1 | 1 | 8 | CA1 | 80 | 4/5 | wt | DZNE |
| AAV1 | Syn.GCaMP6m | 100841 | $1 \times 10^{13}$ | 1:2 | 1 | 6 | CA1 | 43 | 3/7 | wt | DZNE |
| AAV1 | Syn.GCaMP6s | 100843 | $1 \times 10^{13}$ | 1:2 | 0.5 | 4–10 | DG | 0 | 0/3 | wt | UoB |
| AAV9 | Syn.jGCaMP7f | 104488 | $2.5 \times 10^{13}$ | 1:2 | 1 | 10–14 | CA1 | 50 | 3/6 | APPswe[†] | DZNE |
| AAV1 | Syn.GCaMP6s | 100843 | $1 \times 10^{13}$ | 1:1 | 0.25 | 3–12 | CA3 | 100 (1 exp.) | 1/1 | wt | CU |
| AAV1 | Syn.GCaMP6s | 100843 | $1 \times 10^{13}$ | 1:1 | 0.25 | 3–5 | CA1 | 100 (1 exp.) | 1/1 | wt | CU |
| AAV1 | Syn.GCaMP6s | 100843 | $1 \times 10^{13}$ | 1:2 | 0.8 | 4–5 | Ctx | 0 | 0/>20 | wt | CU |
| AAV1 | Syn.GCaMP6f | 100837 | $7 \times 10^{12}$ | 1:2 | 0.75 | 3–6 | Ctx | 0 | 0/>20 | wt | CU |
| AAV9 | hSyn1.R-CaMP1.07 | V224-9 [‡] | $4.3 \times 10^{12}$ | 1:1 | 0.2 | 8 | CA1 | 100 | 2/2 | wt | UZH |
| AAV1 | syn.Flex.GCaMPm | 100838 | $1 \times 10^{13}$ | 1:2 | | | | | | | |
| AAV9 | CamKII0.4.Cre.SV40 | 105558 | $1 \times 10^{13}$ | 1:2 | 0.5 | 34–38 | CA1 | 0 | 0/3 | wt | DZNE |
| AAV1 | syn.Flex.GCaMP6s | 100845 | $1 \times 10^{13}$ | 1:2 | | | | | | | |
| AAV1 | CamKII0.4.Cre.SV40 | 105558 | $1 \times 10^{13}$ | 1:2 | 0.5 | 6 | CA1 | 0 | 0/2 | wt | UoB |

*In heterozygous $Scn2a^{A263V}$ mice.

[†]In PV-Cre::APPswe/PS1dE9 mice.

[‡]Sourced from the Viral Vector Facility University of Zurich (VVF/UZH).

These aberrant $Ca^{2+}$ microwaves robustly occurred and were observed in four different institutes each using a common viral transduction approach and standard two-photon $Ca^{2+}$ imaging protocols.

$Ca^{2+}$ microwaves were typically first detected at ~4 wk, rarely also at 2 wk, after injection. Thus, there may be a time window when $Ca^{2+}$ activity could be recorded in the absence of this artefactual phenomenon. However, we would still hesitate to use this specific approach for hippocampal imaging experiments as, although unknown from our data, more subtle alterations may occur prior to visible onset of aberrant activity. Furthermore, at sites more distal to the injection site with lower expression levels, $Ca^{2+}$ microwaves may not be observed; however, it may very well be that $Ca^{2+}$ microwaves in regions with higher expression will affect fine-scaled neuronal population dynamics in primarily unaffected neighbouring regions.

The presence of $Ca^{2+}$ microwaves was not restricted to a single GCaMP variant or version, and was observed using either GCaMP6m, GCaMP6s, or GCaMP7f. The phenomenon was also observed upon transduction of R-CaMP1.07, indicating that these aberrant hippocampal waves are not restricted to GCaMP indicators, but rather present a general phenomenon following $Ca^{2+}$-indicator transduction. Notably, the viral transduction titre was a key factor as reducing the viral transduction titre from $1 \times 10^{13}$ vc/ml (500 nl or 1000 nl of a 1:1 undiluted virus solution) to $5 \times 10^{12}$ vc/ml (1000 nl 1:2 solution, single injection) decreased, albeit did not yet prevent, the occurrence of $Ca^{2+}$ microwaves. In the literature, hippocampal GCaMP transduction procedures in mice typically include one to several separate nearby injections, with a total volume of transduced undiluted virus ranging from 60 nl to 500 nl (*Cai*

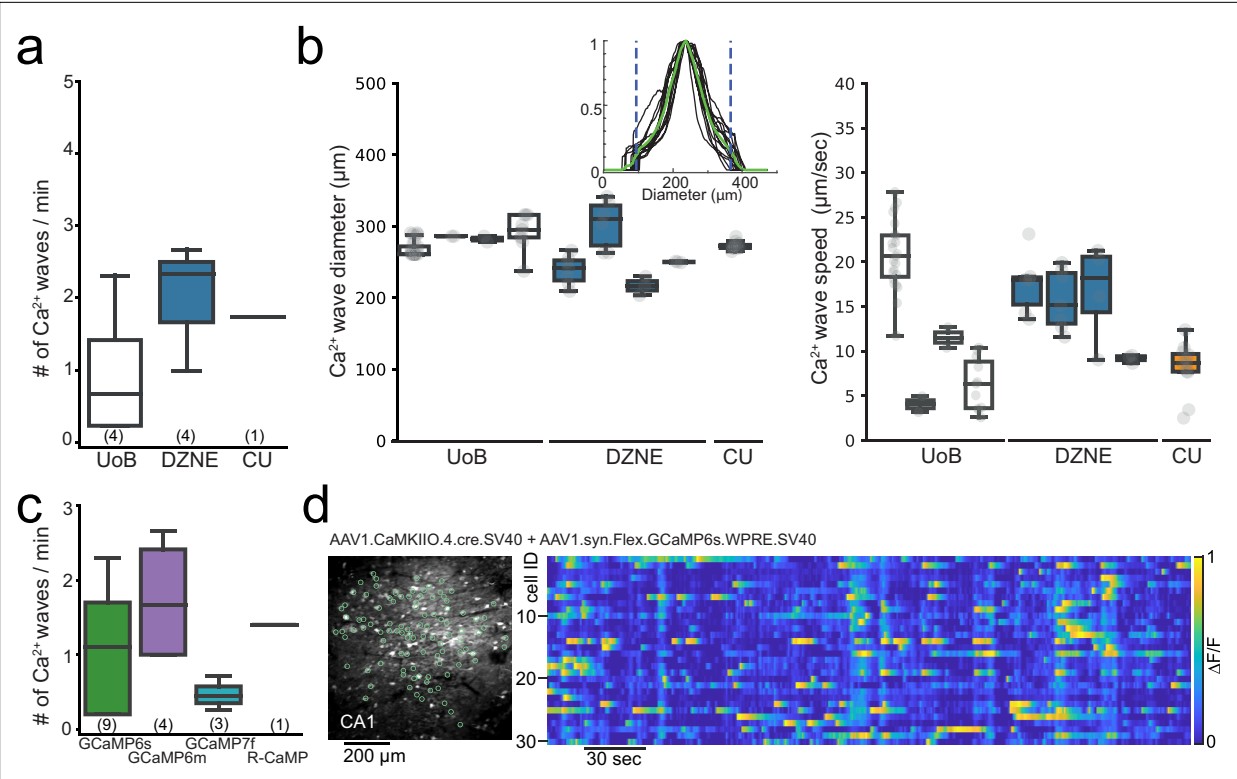

**Figure 2.** Aberrant Ca²⁺ microwaves are consistent across laboratories and genetically encoded calcium indicator (GECI) variant. (**a**) Boxplot (median ± quartiles and range) of the occurrence rate of aberrant Ca²⁺ microwaves in CA1 at the different institutes at 6–8 wk after injection of GCaMP6s or GCaMP6m (number of animals in parenthesis). (**b**) Boxplots (median ± quartiles and range) of Ca²⁺ microwave diameters (left) and progression speed (right) in CA1 from each animal recorded across institutes. Inset: histogram of fluorescent intensity taken across each Ca²⁺-wave within an animal. Green line is the average, areas outside dashed lines mark 10% lowest fluorescence values, which were excluded from analysis. (**c**) Boxlot (median ± quartiles and range) of the occurrence rate of aberrant Ca²⁺ microwaves in CA1 following injection with commonly used GECIs (number of animals in parenthesis; see *Table 1*). (**d**) Two-photon Ca²⁺ imaging field of view (FOV) (left) in hippocampal CA1 following dual injection approach for conditional GCaMP6s expression (6 wk post-injection [p.i.]) with normal sparse spontaneous Ca²⁺ activity and no detection of Ca²⁺ microwaves (right; raster plot of ΔF/F, 1 min moving window, traces max-normalized per neuron).

*et al., 2016*; *Jimenez et al., 2020*; *Keinath et al., 2022*; *Pettit et al., 2022*; *Radvansky et al., 2021*; *Skocek et al., 2018*; *Szabo et al., 2022*; *Weisenburger et al., 2019*; *Wirtshafter and Disterhoft, 2022*; *Zaremba et al., 2017*). In other studies, syn-GCaMP virus was diluted prior to injection (up to 1:10) (*Jimenez et al., 2020*; *Zong et al., 2022*), resulting in varied transduction volumes up to 1500 nl. In our hands, a reduced viral titre of $5 \times 10^{12}$ vc/ml in a 1000 nl injection volume still resulted in aberrant Ca²⁺ microwaves. Thus, viral transduction titres per volume well below this number and diluted transduction solutions are advisable for syn-GCaMP expression in the hippocampus if AAV syn-Ca²⁺-indicator transduction is desired for a planned in vivo hippocampal imaging experiment.

If possible, alternate viral GCaMP expression approaches should be chosen. As a possible alternative, similar to previous reports using dual AAV injections or AAV in Cre-driver mouse lines (*Farrell et al., 2020*; *Grosmark et al., 2021*; *Mineur et al., 2022*; *Rolotti et al., 2022*; *Terada et al., 2022*), we find that Cre-dependent AAV GCaMP expression (IEECR/UoB) in pyramidal neurons does not cause aberrant hippocampal Ca²⁺ microwaves. Moreover, we have not observed this aberrant phenomenon in transgenic thy1-GCaMP6s or 6f mice (JAX strain 025776 or 024276) (*Masala et al., 2023*; *Rupprecht et al., 2023*; *Wenzel et al., 2019b*), nor in Vglut1-IRES2-Cre-D × Ai162(TIT2L-GC6s-ICL-tTA2)-D mice (JAX strains 037512, 031562). It goes beyond the scope and available resources in our laboratories to further identify which viral GCaMP transduction approaches avoid the reported phenomenon. It seems likely that the underlying mechanisms for this artefact comprise transduction titre and time period of GCaMP or R-CaMP1.07 expression, region specificity, and density of expression. Importantly, although our data suggest some regions and AAV constructs seem more prone to generate

artefactual $Ca^{2+}$ waves under certain conditions, this does not mean that $Ca^{2+}$ waves cannot be generated in other regions or with other constructs or promoters. It remains unclear whether the observed phenomenon is restricted to $Ca^{2+}$ indicator viral expression in mice or whether it extends to different animal models as well. In this regard, a previous report did not observe $Ca^{2+}$-waves in rats following synapsin-dependent GCaMP6m expression, although notably, imaging was performed under isoflurane anaesthesia (*Sosulina et al., 2021*). Furthermore, disentangling the exact cellular mechanisms of the phenomenon from technical aspects is difficult as, for example, the mere change in the transduction procedure will affect GECI expression level. For instance, although $Ca^{2+}$ waves were not observed following conditional expression of GCaMP with CaMKII.Cre, which may suggest a requirement for interneuronal expression, it may also simply reflect differences in final GCaMP expression density and levels between the two transduction procedures.

In the context of this study, the phenomenon of $Ca^{2+}$ microwaves is possibly related to the expression of exogenous $Ca^{2+}$ buffer and the resulting effects on $Ca^{2+}$ dynamics and gene expression (*McMahon and Jackson, 2018*; *Rose et al., 2014*; *Yang et al., 2018*), which may be why our findings extended across genetically encoded $Ca^{2+}$ indicators. Beyond clearly being abnormal, the exact nature of the observed $Ca^{2+}$ microwaves remains unclear and may reflect $Ca^{2+}$ influx during action potential firing or possibly $Ca^{2+}$ release from internal stores. In a limited dataset, we tried to detect the $Ca^{2+}$ microwaves by hippocampal LFP recordings (insulated tungsten wire, diameter ~110 μm). We could not identify a specific signature, for example, ictal activity or LFP depression, which may correspond to these $Ca^{2+}$ microwaves. The shortcoming of these LFP recordings is that we could not simultaneously perform hippocampal two-photon microscopy, and thus, it is uncertain whether the $Ca^{2+}$ microwaves indeed occurred in proximity to our electrode. We did not evaluate the effect of $Ca^{2+}$ microwaves on physiological activity. Based on the data presented here, it appears reasonable to hypothesize that such waves obscure if not interfere with physiological activity, for example, with hippocampal place cell activity. However, the primary purpose of this article was to inform the community about an artefact that can be avoided using alternative approaches.

In summary, this report shows that common AAV hippocampal injection procedures of $Ca^{2+}$ indicators may lead to aberrant $Ca^{2+}$ microwaves in wildtype mice and genetic mouse models of disease, particularly if high-titre virus loads are used. The aim of this article is not to discredit $Ca^{2+}$ indicators expressed under the synapsin promoter, a tool that we greatly appreciate ourselves, but to sensitize the field to artefactual transduction-induced aberrant $Ca^{2+}$ microwaves. The underlying mechanisms, some of which we have described above, are likely multifaceted. This article seeks to inform and alert others to carefully evaluate their $Ca^{2+}$ indicator expression approach for in vivo $Ca^{2+}$ imaging of the hippocampus, which is becoming increasingly popular. There is certainly a much greater number of safe alternate hippocampal $Ca^{2+}$ indicator viral expression approaches than has been reported here, and we encourage others to report on viral $Ca^{2+}$ indicator transduction safety profiles. Indeed, others have also encountered these artefactual events as recent social media posts attest (*Application Specialist Team, 2023*). With more indicators of brain cell activity becoming available ($Ca^{2+}$ indicators and others including voltage indicators) as well as routes for viral delivery (*Grødem et al., 2023*), the open and timely reporting of transduction safety profiles will reduce unnecessary animal experiments and save laboratory resources and time in future investigations into hippocampal function in health and disease.

## Materials and methods

### Animals

All experiments followed the EU animal welfare law (University of Bonn [81-02.04.2019.A139, 81-02.04.2019.A288], DZNE [84-02.04.2013.A356, 81-02.04.2018.A063]) or institutional guidelines of the Animal Care and Use Committee and respective federal office (Columbia University [AC-AAAV3464, AC-AAAM8851, AC-AAAH1804], University of Zurich [ZH211/2018]). We used wildtype C57BL/6J mice, Thy1-GCaMP6 mice (C57BL/6J-Tg(Thy1-GCaMP6s)GP4.12Dkim/J; Jackson Lab stock no. 025776, or C57BL/6J-Tg(Thy1-GCaMP6f)GP5.5Dkim/J; Jackson Lab stock Nno. 024276 [*Dana et al., 2019*]), Vglut1-IRES2-Cre-D mice (Jackson Lab stock no. 037512) crossed with Ai162(TIT2L-GC6s-ICL-tTA2)-D mice (Jackson Lab stock no. 031562), PV-Cre::APPswe/PS1dE9 (cross between B6;129P2-Pvalbtm1(cre)Arbr/J, Jackson Lab stock no. 008069, and B6.Cg-Tg(APPswe,PSEN1dE9)85Dbo/

**Table 2.** Viral constructs used.

Resources table

| Genetic reagent (*Mus musculus*) | Recombinant DNA reagent | AAV | Source | ID |
|---|---|---|---|---|
| Syn.GCaMP6s | Syn.GCaMP6s.WPRE.SV40 | AAV1 | Addgene | 100843 |
| Syn.GCaMP6m | Syn.GCaMP6m.WPRE.SV40 | AAV1 | Addgene | 100841 |
| Syn.jGCaMP7f | Syn-jGCaMP7f-WPRE | AAV9 | Addgene | 104488 |
| Syn.GCaMP6f | Syn.GCaMP6f.WPRE.SV40 | AAV1 | Addgene | 100837 |
| hSyn1.R-CaMP1.07 | hSyn1-chl-RCaMP1.07-WPRE-SV40p(A) | AAV9 | VVF/UZH | V224-9 |
| syn.Flex.GCaMP6s | Syn.Flex.GCaMP6s.WPRE.SV40 | AAV1 | Addgene | 100845 |
| syn.Flex.GCaMP6m | Syn.Flex.GCaMP6m.WPRE.SV40 | AAV1 | Addgene | 100838 |
| CamKII0.4.Cre.SV40 | CamKII 0.4.Cre.SV40 | AAV1 or 9 | Addgene | 105558 |

Mmjax, Jackson Lab stock no. 034832) or *Scn2a*[A263V] mice (from *Schattling et al., 2016*). Mice were kept under a light schedule of 12 hr on/12 hr off, constant temperature of 22 ± 2°C, and humidity of 65%. They had ad libitum access to water and standard laboratory food at all times. All efforts were made to minimize animal suffering and reduce the number of animals used.

## Virus injections

For in vivo two-photon imaging experiments, GECIs were virally transduced using injection of an AAV (see *Tables 1* and *2*). At the time of injection, mice ranged in age from 5 to 79 wk. There was no significant relationship between the age of the animal and the incidence nor frequency of $Ca^{2+}$ microwaves during this period (linear regression fit to the $Ca^{2+}$ wave frequency against age was not significant: intercept = 1.37, slope = –0.007, p=0.62, n = 14; and generalized linear model relating $Ca^{2+}$ wave incidence ~ age was not significant: z score = 0.19, deviance above null = 0.04, p=0.85, n = 24).

### At IEECR/University of Bonn

Mice (~6 wk of age) received ketoprofen (Gabrilen, Mibe; 5 mg/kg body weight [b.w.]; injection volume 0.1 ml/10 g b.w., subcutaneously [s.c.]) for analgesia and anti-inflammatory treatment 30 min prior to induction of anaesthesia. Then, mice were anaesthetized with 2–3% isoflurane in an oxygen/ air mixture (25/75%) and then placed in a stereotactic frame. Eyes were covered with eye-ointment (Bepanthen, Bayer) to prevent drying, and body temperature was maintained at 37°C using a regulated heating plate (TCAT-2LV, Physitemp) and a rectal thermal probe. After hair removal and superficial disinfection, a drop of 10% lidocaine was used to locally anaesthetize the area. After 3–5 min, a flap of skin was removed about 1 cm² around the middle of the skull. Residual soft tissue was then removed from the skull with a scraper and 3% $H_2O2$/NaCl solution. After complete drying, cranial sutures served as landmarks for the determination of injection sites. For virus injection, a burr hole was carefully drilled through the skull using a dental drill, avoiding excessive heating and injury to the meninges by intermittent cooling with sterile PBS. Coordinates were, for CA1, anterioposterior (AP) measured from bregma 1.9 mm, lateral (L) specified from midline 1.6 mm, dorsoventral (DV) from the surface of the skull 1.6 mm; for DG, AP 2.4 mm; L 1.6 mm; ×3 injections at DV 2.7, 2.5, and 2.1 mm. Virus particles (see *Table 1*) were slowly injected (20–100 nl/min). To prevent reflux of the injected fluid upon cannula retraction, it was left in place until 5 min post-injection and then carefully lifted.

### At DZNE

A more detailed procedure was described previously (*Fuhrmann et al., 2015*; *Poll et al., 2020*). Briefly, mice (6–78 wk) were anaesthetized with i.p. injection of ketamine (0.13 mg/g) and xylazine (0.01 mg/g), head-fixed using a head holder (MA-6N, Narishige, Tokyo, Japan) and placed into a motorized stereotactic frame (Luigs-Neumann, Ratingen, Germany). Body temperature was constantly controlled by a self-regulating heating pad (Fine Science Tools, Heidelberg, Germany). After skin incision and removal of the pericranium, the position of the injection 34 G cannula was determined in relation to bregma. A 0.5 mm hole was drilled through the skull (Ideal Micro Drill, World Precision

Instruments, Berlin, Germany). Stereotactic coordinates were taken from Franklin and Paxinos, 2008 (The Mouse Brain in Stereotaxic Coordinates, Third Edition, Academic Press). Virus (see *Table 1*) was injected in two loci with the following CA1 coordinates: AP 1.95 mm; L 1.5 mm; DV 1.15 mm at a speed of 100 nl/min.

### At Columbia University

A more detailed procedure was described previously (*Wenzel et al., 2017*; *Wenzel et al., 2019a*). Briefly, mice (8–20 wk) were anaesthetized with isoflurane (initial dose 2–3% partial pressure in air, then reduction to 1–1.5%). For viral injections, a small cranial aperture was established using a dental drill above the somatosensory cortex (coordinates from bregma: AP 2.5 mm, L 0.24 mm, DV 0.2 mm), or V1 (coordinates from lambda: AP 2.5 mm, L 0.02 mm, DV 0.2–0.3 mm), or the hippocampus (coordinates from bregma, CA1: –1.9 mm, –1.6 mm, –1.6 mm; CA3: –2.2 mm, –2.3 mm, –2.7 mm). A glass capillary pulled to a sharp micropipette was advanced with the stereotaxic instrument, and virus particles (see *Table 1*) were injected into putative layer 2/3 of neocortex over a 5 min period at 50 nl/min, or hippocampus over 12.5 min using a UMP3 microsyringe pump (World Precision Instruments).

### At University of Zurich

A more detailed procedure was described previously (*Rupprecht et al., 2023*). Briefly, mice (18 wk) were anaesthetized using isoflurane (5% in $O_2$ for induction, 1–2% for maintenance during surgery) and provided with analgesia (Metacam 5 mg/kg b.w., s.c.). Body temperature was maintained at 35–37°C using a heating pad. An incision was made into the skin after local application of lidocaine. Viral particles (see *Table 1*) were injected into CA1 (coordinates AP –2.0 mm, ML –1.5 mm from bregma, DV –1.3) using a glass pipette with a manually driven syringe at a rate of approximately 50 nl/min. The injection pipette was left in place for further 5 min before being slowly retracted.

## In vivo imaging window implantation procedure

Cranial window surgery was performed to allow imaging from the dorsal hippocampal CA1/CA3 region or neocortex.

### At IEECR, University of Bonn

Thirty minutes prior to induction of anaesthesia, buprenorphine was administered for analgesia (Buprenovet, Bayer; 0.05 mg/kg b.w.; injection volume 0.1 ml/20 g b.w., intraperitoneally [i.p.]). Furthermore, dexamethasone (Dexa, Jenapharm; 0.1 mg/20 g b.w.; injection volume 0.1 ml/20 g b.w., i.p.), and ketoprofen (Gabrilen, Mibe; 5 mg/kg b.w.; injection volume 0.1 ml/10 g b.w., s.c.) were applied to counteract inflammation, swelling, and pain. Mice were anaesthetized with 2–3% isoflurane in an oxygen/air mixture (25/75%) and then placed in a stereotactic frame. Eyes were covered with eye-ointment (Bepanthen, Bayer), and body temperature was maintained at 37°C by closed-loop regulation through a warming pad (TCAT-2LV, Physitemp) and a rectal thermal probe. Throughout the course of the surgical procedure, the isoflurane dose was successively reduced to about 1–1.5% at a gas flow rate of ~0.5 ml/min. A circular craniotomy (Ø ~ 3 mm) was established above the right hemisphere/hippocampus within the central opening (Ø ~ 7 mm) of the head plate using a dental drill. Cortical tissue was carefully aspirated until the alveolar fibres above CA1 could be visually identified. A custom-made silicon cone (top Ø 3 mm, bottom Ø 2 mm, depth 1.5 mm, RTV 615, Momentive) attached to a cover glass (Ø 5 mm, thickness 0.17 mm) was inserted and fixed with dental cement around the edges of the cover glass (see *Masala et al., 2023*). Postoperatively, all mice received analgetic treatment by administration of buprenorphine twice daily (Buprenovet, Bayer; 0.05 mg/kg b.w.; injection volume 0.1 ml/20 g b.w., i.p.) and ketoprofen once daily (Gabrilen, Mibe; 5 mg/kg b.w.; injection volume 0.1 ml/10 g b.w., s.c.) for three consecutive days post-surgery. Throughout this time, animals were carefully monitored twice daily. Animals typically recovered from surgery within 24–48 hr, showing normal activity and no signs of pain or distress.

### At DZNE

Prior to surgery, mice were anaesthetized with an intraperitoneal injection of ketamine/xylazine (0.13/0.01 mg per gram of body weight). Additionally, an anti-inflammatory (dexamethasone, 0.2 mg/

kg) and an analgesic drug (buprenorphine hydrochloride, 0.05 mg/kg; Temgesic, Reckitt Benckiser Healthcare) were subcutaneously administered. A cranial window (Ø 3 mm) was implanted above the right hippocampus as previously described (*Poll et al., 2020*).

### At Columbia University

For neocortical imaging, directly following virus injection, the craniotomy was covered with a thin glass cover slip (3 × 3 mm, No. 0, Warner Instruments), which was fixed in place with a slim meniscus of silicon around the edge of the glass cover and finally cemented on the skull using small amounts of dental cement around the edge. For hippocampal imaging, a small area of cortex (around 1.5 × 1.5 mm) above the left CA1 was removed by gentle suction down to the external capsule, as described previously (*Dombeck et al., 2010*; *Wenzel et al., 2019b*). The site was repeatedly rinsed with sterile saline until no further bleeding could be observed. Then, a small UV-sterilized miniature glass plug (1.5 × 1.5 mm, BK7 glass, obtained from BMV Optical), glued to the centre of a thin glass coverslip (3 × 3 mm, No. 0, Warner Instruments) with UV-sensitive glue, was carefully lowered onto the external capsule until the edges of the attached glass cover touched the skull surrounding the craniotomy. Finally, the plug was fixed in place with a slim meniscus of silicon around the edge of the glass cover and by applying small amounts of dental cement around the edge of the glass cover.

### At University of Zurich

A more detailed procedure was described previously (*Rupprecht et al., 2023*). Briefly, 2 wk after virus injection, the hippocampal window was implanted. Two layers of light-curing adhesive (iBond Total Etch, Kulzer) were applied to the exposed skull, followed by a ring of dental cement (Charisma, Kulzer). A 3-mm-diameter ring was drilled into the skull, centred at the previous injection site. The cortex in the exposed region was carefully aspirated using a vacuum pump until the stripes of the corpus callosum became visible. The corpus callosum was left intact. A cylindrical metal cannula (diameter 3 mm, height 1.2–1.3 mm) attached with dental cement to a coverslip (diameter 3 mm) was carefully inserted into the cavity. The hippocampal window was fixed in place using UV-curable dental cement (Tetric EvoFlow, Ivoclar).

## Two-photon Ca²⁺ imaging

A variety of standard commercially available two-photon systems were used at the different institutes to record the $Ca^{2+}$ microwaves.

### At IEECR, University of Bonn

A commercially available two-photon microscope was used (A1 MP, Nikon), equipped with a ×16 water-immersion objective (N.A. = 0.8, WD = 3 mm, CFI75 LWD 16X W, Nikon), and controlled using NIS-Elements software (Nikon). GCaMP6s was excited at 940 nm using a Ti:sapphire laser system (~60 fs laser pulse width; Chameleon Vision-S, Coherent). Emitted photons were collected using gated GaAsP photomultipliers (H11706-40, Hamamatsu). Several individual tif series were recorded by resonant scanning at a frame rate of 15 Hz for a total 20–40 min per imaging session.

### At DZNE

Recordings of $Ca^{2+}$-changes were performed with a galvo-resonant scanner (Thorlabs, Newton, USA) on a two-photon microscope equipped with a ×16 water immersion objective with a numerical aperture of 0.8 (N16XLWD-PF, Nikon, Düsseldorf, Germany) and a titanium sapphire (Ti:Sa) 80 MHz Cameleon Ultra II two-photon laser (Coherent, Dieburg, Germany) that was tuned to 920 nm for GCaMP6m fluorescence excitation. GCaMP6m fluorescence emission was detected using a band-pass filter (525/50 nm, AHF, Tübingen, Germany) and a GaAsP PMT (Thorlabs). ThorImageLS software (Thorlabs, version 2.1) was used to control image acquisition. Image series (896 × 480 pixels, 0.715 µm/pixel, or 640 × 256 pixels) were acquired at 30.3 Hz or 32.3 Hz.

### At Columbia University

Neural population activity was recorded using a commercially available two-photon microscope (Bruker; Billerica, MA) and a Ti:sapphire laser (Chameleon Ultra II; Coherent) at 940 nm through a ×25

objective (Olympus, water immersion, N.A. 1.05). Resonant galvanometer scanning and image acquisition (frame rate 30.206 fps, 512 × 512 pixels) were controlled using Prairie View Imaging software.

### At University of Zurich

Neuronal population activity was recorded using a custom-built two-photon microscope (see *Rupprecht et al., 2023*). Briefly light from a femtosecond-pulsed laser (MaiTai, Spectra Physics; tuned to 960 nm; power below objective 40–45 mW) was used to scan the sample below a ×16 objective (Nikon, water immersion, NA 0.8). Image acquisition and scanning (frame rate 30.88 Hz, 622 × 512 pixels) were controlled using custom-written software (*Chen et al., 2013*).

## Analysis of aberrant Ca$^{2+}$ microwaves

To remove motion artefacts, recorded movies were registered using a Lucas–Kanade model (*Greenberg and Kerr, 2009*) or the ImageJ plugin moco (available through the Yuste web page or https://github.com/NTCColumbia/moco, copy archived by *NTCColumbia, 2016*; *Dubbs et al., 2016*), or in the case of R-CaMP the NoRMCorre algorithm (*Pnevmatikakis and Giovannucci, 2017*).

We determined the diameter of the calcium waves in a semi-automated fashion from the raw tif series. Using ImageJ software, we first drew an orthogonal line across the largest aspect of each calcium wave progressing through the FOV, which resulted in a fluorescent histogram for each wave. Using custom code (MATLAB R2020b), we further analysed all histograms for each mouse and imaging time point. First, we applied a gentle smoothing, max-normalized each histogram, and max-aligned all histograms of a given imaging session. Then, after excluding the 10% lowest fluorescent values, the width of each calcium wave and a mean value were calculated for each time point/imaging session. Finally, the resulting pixel values were converted to micrometer, based on the respective objective (@ 512 × 512 pixels and ×1 zoom: Nikon ×16, NA 0.8, 3 mm WD: 1.579 µm/pixel; Olympus ×25, NA 1.05 2 mm WD: 0.92 µm/pixel). The speed of the Ca$^{2+}$ microwaves was calculated from the duration and path length of the events visually identified and manually tracked in the FOV.

## Histochemistry

To verify successful viral transduction and window position, animals were deeply anaesthetized with ketamine (80 mg/kg b.w.) and xylazine (15 mg/kg b.w.) at the end of the respective experiment. After confirming a sufficient depth of anaesthesia, mice were heart-perfused with cold phosphate-buffered saline (PBS) followed by 4% paraformaldehyde (PFA) in PBS. Animals were decapitated and the brain removed and stored in 4% PFA in PBS solution for 24 hr. Then, 50–100-µm-thick coronal slices of the hippocampus were cut on a vibratome (Leica). For nuclear staining, brain slices were kept for 10 min in a 1:1000 DAPI solution at room temperature. Brain slices were mounted and the green and blue channel successively imaged under an epi-fluorescence or spinning disc microscope (Visitron VisiScope).

## Acknowledgements

We thank Lea Adenauer, Laura Kück, and Olga Zabashta for excellent technical support with animal husbandry and immunohistochemistry. We acknowledge the support of the Imaging Core Facility of the Bonn Technology Campus Life Sciences funded by the Deutsche Forschungsgemeinschaft (DFG, German Research Foundation) – Projektnummer 388169927. The work was supported by the Deutsche Forschungsgemeinschaft (DFG, German Research Foundation) with a Research Group FOR-2715 to TK and HB, SPP2395 to MF and HB, SFB1089 to MF, MW, and HB, and BE 1822/13-1 to HB. The work was further supported by the BONFOR Program UoB (MW: #2019-2-04), the Hertie Network of Excellence in Clinical Neuroscience (MW: #P1200008), European Research Council (MW: StG #101039945; FH: AdvG #670757, MF: CoG#865618), the Swiss National Science Foundation (project grant 310030B_170269 and Sinergia grant CRSII5 180316 to FH; Ambizione grant PZ00P3_209114 to PR). The work was further supported by the iBehave network (MF, HB, MW). Additional support was provided by the National Science Foundation (NSF GFRP to DAO; 2203119 to RY), National Institute of Neurological Disorders and Stroke (F99NS134209 to DAO and RM1NS132981 to RY), National Institute of Mental Health (R01MH115900 to RY), NSF (2203119 to RY), and a Vannevar Bush Faculty Fellowship (ONR N000142012828 to RY).

# Additional information

## Funding

| Funder | Grant reference number | Author |
|---|---|---|
| Deutsche Forschungsgemeinschaft | FOR-2715 | Heinz Beck Tony Kelly |
| Deutsche Forschungsgemeinschaft | SPP2395 | Martin Fuhrmann Heinz Beck |
| Deutsche Forschungsgemeinschaft | SFB1089 | Martin Fuhrmann Heinz Beck Michael Wenzel |
| Deutsche Forschungsgemeinschaft | BE 1822/13-1 | Heinz Beck |
| European Research Council | | Fritjof Helmchen Martin Fuhrmann Michael Wenzel |
| National Science Foundation | | Darik A O'Neil Rafael Yuste |

The funders had no role in study design, data collection and interpretation, or the decision to submit the work for publication.

## Author contributions

Nicola Masala, Manuel Mittag, Eleonora Ambrad Giovannetti, Darik A O'Neil, Fabian J Distler, Investigation, Writing – review and editing; Peter Rupprecht, Funding acquisition, Investigation, Writing – review and editing; Fritjof Helmchen, Rafael Yuste, Martin Fuhrmann, Resources, Supervision, Funding acquisition; Heinz Beck, Resources, Supervision, Funding acquisition, Writing – review and editing; Michael Wenzel, Conceptualization, Formal analysis, Funding acquisition, Investigation, Writing – original draft, Writing – review and editing; Tony Kelly, Conceptualization, Resources, Formal analysis, Supervision, Funding acquisition, Writing – original draft, Writing – review and editing

## Author ORCIDs

Peter Rupprecht ⓘ https://orcid.org/0000-0001-8235-8257
Fritjof Helmchen ⓘ https://orcid.org/0000-0002-8867-9569
Rafael Yuste ⓘ https://orcid.org/0000-0003-4206-497X
Martin Fuhrmann ⓘ https://orcid.org/0000-0001-7672-2913
Heinz Beck ⓘ https://orcid.org/0000-0002-8961-998X
Michael Wenzel ⓘ https://orcid.org/0000-0002-6065-1660
Tony Kelly ⓘ https://orcid.org/0000-0001-6066-0455

## Ethics

All experiments followed the EU animal welfare law (University of Bonn [81-02.04.2019.A139, 81-02.04.2019.A288], DZNE [84-02.04.2013.A356, 81-02.04.2018.A063]) or institutional guidelines of the Animal Care and Use committee and respective Federal office (Columbia University [AC-AAAV3464, AC-AAAM8851, AC-AAAH1804], University of Zurich [ZH211/2018]).

Reviewer #2 (Public Review): https://doi.org/10.7554/eLife.93804.3.sa1
Reviewer #3 (Public Review): https://doi.org/10.7554/eLife.93804.3.sa2
Reviewer #4 (Public Review): https://doi.org/10.7554/eLife.93804.3.sa3
Author response https://doi.org/10.7554/eLife.93804.3.sa4

# Additional files

## Supplementary files
• MDAR checklist

## Data availability

Many example videos are included in the manuscript and supporting files. Raw data from a subset of animals are available at Zenodo (https://doi.org/10.5281/zenodo.12655766), due to size restrictions the full videos are available from the corresponding author upon request. Data and code required to reproduce the summary data figures is available at (https://github.com/tonykelly00/Ca_waves_data-Elife, copy archived at *Kelly, 2024*).

The following dataset was generated:

| Author(s) | Year | Dataset title | Dataset URL | Database and Identifier |
|-----------|------|---------------|-------------|-------------------------|
| Kelly T | 2024 | In-vivo two-photon imaging of aberrant Ca2+-waves following viral transduction of Ca2+ indicators in mice | https://doi.org/10.5281/zenodo.12655766 | Zenodo, 10.5281/zenodo.12655766 |

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
