## [Editor Report · eLife assessment]

This **important** study provides **convincing** evidence of artefactual calcium microwaves during calcium imaging of populations of neurons in the hippocampus using methods that are common in the field. The work raises awareness of these artefacts so that any research labs planning to do calcium imaging in the hippocampus can avoid them by using alternative strategies that the authors propose.

---

## [Referee Report · Reviewer #2 (Public Review)]

Summary:

The authors describe and quantify a phenomenon in the CA1 and CA3 of the hippocampus that they call aberrant Ca2+ micro-waves. Micro-waves are sometimes seen during 2-photon calcium imaging of populations of neurons under certain conditions. They are spatially confined slow calcium events that start in a few cells and slowly spread to neighboring groups of cells. This phenomenon has been uttered between researchers in the field at conferences, but no one has taken the time to carefully capture and quantify micro-waves and pin down the causes. The authors show that micro-waves are dependent on the viral titre of the genetically encoded calcium indicators (GECIs), the genetic promoter (synapsin), the neuronal subtype (granule cells in the dentate gyrus do not produce micro-waves and they are not seen in the neocortex), and the density of GECI expression. The authors should be commended for their work and for raising awareness to all labs doing any form of calcium imaging in populations of neurons. The authors also come up with alternative approaches to avoid artifactual micro-waves such as reducing the transduction titre (1:2 dilution of virus) and a transduction method employing sparser and cre-dependent GECI expression in principal cells using a CaMKII promoter.

Strengths:

The micro-waves reported in the paper were robustly observed across 4 laboratories and 3 different countries with various experimenters and calcium imaging set-ups. This adds significant strength to the work.

The age of mice used covered a broad range (from 6 to 43 weeks). This is a strength because it covers most ages that are used in labs that regularly do calcium imaging.

Another strength is they used different GCaMP variants (GCaMP6m, GCaMP6s, GCaMP7f), as well as a red indicator: RCaMP. This shows the micro-waves are not an issue with any particular GECI, as the authors suggest.

The authors include many movies of micro-waves. This is extremely useful for researchers in the field to view them in real-time so they can identify them in their own data.

They provide a useful table with specific details of the virus injected, titre, dilution, and other information along with the incidence of micro-waves. A nice look-up table for researchers to see if their viral strategy is associated with a high or low incidence of micro-waves.

Weaknesses:

The effect of mico-waves on single cell function was not analyzed. It would be useful, for example, if we knew the influence of micro-waves on place fields. Can a place cell still express a place field in a hippocampus that produces micro-waves? What effect might a microwave passing over a cell have on its place field? Mice were not trained in these experiments, so the authors do not have the data. However, they do briefly discuss these ideas.

---

## [Referee Report · Reviewer #3 (Public Review)]

Summary:

The work by Masala and colleagues highlights a striking artifact that can result from a particular viral method for expressing genetically encoded calcium indicators (GECIs) in neurons. In a cross-institutional collaboration, the authors find that viral transduction of GECIs in the hippocampus can result in aberrant slow-traveling calcium (Ca2+) micro-waves. These Ca2+ micro-waves are distinct from previously described ictal activity but nevertheless are likely a pathological consequence of overexpression of virally transduced proteins. Ca2+ micro-waves will most-likely obscure the physiology that most researchers are interested in studying with GECIs, and their presence indicates that the neural circuit is in an unintended pathological state. Interestingly this pathology was not observed using the same viral transduction methods in other brain regions. The authors recommend several approaches that may help other experimenters avoid this confound in their own data such as reducing the titer of viral injections or using recombinase-dependent expression. The intent of this manuscript is to raise awareness of the potential unintended consequences of viral overexpression, particularly for GECIs. A rigorous investigation into the exact causes of Ca2+ micro-waves or the mechanisms supporting them are beyond the authors' intended scope.

Strengths:

The authors clearly demonstrate that Ca2+ micro-waves occur in the CA1 and CA3 regions of the hippocampus following large volume, high titer injections of adeno-associated viruses (AAV1 and AAV9) encoding GECIs. The supplementary videos provide undeniable proof of their existence.

By forming an inter-institutional collaboration, the authors demonstrate that this phenomenon is robust to changes in surgical techniques or imaging conditions.

Weaknesses:

I believe that the weaknesses of the manuscript are appropriately highlighted by the authors themselves in the discussion. The manuscript does not attempt to exhaustively characterize the conditions under which calcium micro-waves occur. Rather, the authors raise awareness of this problem.

---

## [Referee Report · Reviewer #4 (Public Review)]

Summary:

Masala N et al showed interesting aberrant calcium microwaves in the hippocampus when synapsin promoter driven GCaMPs were expressed for a long period of time. These aberrant hippocampal Ca2+ micro-waves depend on the viral titre of the GECI. The microwave of Ca2+ was not observed when GECI was expressed only a sparse set of neurons.

Strengths:

These findings are important to wide neuroscience community especially when considering a great number of investigators are using similar approaches. Results look convincing and are consistent across several laboratories.

Weaknesses:

Synapsin promoter labels both excitatory pyramidal neurons and inhibitory neurons. To avoid aberrant Ca2+ microwave, a combination of Flex virus and CaMKII-Cre or Thy-1-GCaMP6s and 6f mice were tested. However, all these approaches limit the number of infected pyramidal neurons. While the comprehensive display of these results is appreciated, one additional important test would be more informative. To distinguish whether the microwave of Ca2+ is sufficiently caused via the expression of GCaMP in interneurons, or just a matter of pyramidal neuron density, testing Flex-GCaMP6 in interneuron specific mouse lines such as PV-Cre and SOM-Cre will provide further clarifications.

---

## [Author Response]

The following is the authors’ response to the original reviews.

**Public Reviews:**

**Reviewer #1 (Public Review):**
Weaknesses:One important question is needed to further clarify the mechanisms of aberrant Ca2+ microwaves as described below.Synapsin promoter labels both excitatory pyramidal neurons and inhibitory neurons. To avoid aberrant Ca2+ microwave, a combination of Flex virus and CaMKII-Cre or Thy-1-GCaMP6s and 6f mice were tested. However, all these approaches limit the number of infected pyramidal neurons. While the comprehensive display of these results is appreciated, a crucial question remains unanswered. To distinguish whether the microwave of Ca2+ is caused selectively via the abnormality of interneurons, or just a matter of pyramidal neuron density, testing Flex-GCaMP6 in interneuron specific mouse lines such as PV-Cre and SOM-Cre will be critical.

We agree that unravelling the role of interneurons is important to the understanding of the cellular mechanisms. However, the primary goal of this preprint was to alert the field and those embarking on in vivo Ca2+ imaging to AAV transduction induced artefacts mediated by one of the most widely used viral constructs for Ca2+ imaging in the field. It was important to us to distribute this finding among the community in a timely manner to avoid the unnecessary waste of resources.

We consider a thorough understanding of cell-type specific mechanisms interesting. However, the biological relevance of the Ca2+ waves is as yet unclear and to disentangle exactly which cellular and subcellular factors that drive the aberrant phenomenon will require a large systematic effort which goes beyond our resources. For instance, it will be technically not trivial to separate biologically relevant contributions from technical differences. For instance, the absence of Ca2+ waves under the principal neuron promotor CaMKII may suggest the involvement of interneurons. However, alternate possibilities are a reduced density of expression across principal neurons or that the expression levels between the 2 promoters is different.

The important, take-home message of the preprint, in our opinion, is that users check carefully their viral protocols, adjust the protocols for their specific scientific question and report any issues. We now emphasise the fact that although Ca2+ waves were not observed following conditional expression of syn.GCaMP with CaMKII.cre, this may not be due to a requirement for interneuronal expression but simply reflect differences in final GCaMP expression density and levels between the two transduction procedures (P12, L298-303).

**Reviewer #2 (Public Review):**
Weaknesses:Whether micro-waves are associated with the age of mice was not quantified. This would be good to know and the authors do have this data.

We plotted the animal age at the time of injection for all injections of Syn.GCaMP6 into CA1/CA3 and found no correlation in either the occurrence of Ca2+ waves nor the frequency of Ca2+ waves during the age period between 5 – 79 wks (see reviewer Fig1; linear regression fit to the Ca2+ wave frequency against age was not significant: intercept = 1.37, slope = -0.007, p=0.62, n = 14; and generalized linear model relating Ca2+ wave ~ age was not significant: z score = 0.19, deviance above null = 0.04, p = 0.85, n = 24). We have now added a statement to this in the revised manuscript (P14 L354-359) and for the reviewers we have added the plots below.

**Author response image 1. sa4fig1:** Plot of Ca2+ micro-wave frequency (left: number of Ca2+ waves/min) or occurrence (right: yes/no) against the animal age at the time of viral injection. Blue line is linear (left) or logistic (right) fit to the data with 95% confidence level.

The effect of micro-waves on single cell function was not analyzed. It would be useful, for example, if we knew the influence of micro-waves on place fields. Can a place cell still express a place field in a hippocampus that produces micro-waves? What effect might a microwave passing over a cell have on its place field? Mice were not trained in these experiments, so the authors do not have the data.

We agree that these are interesting questions; however, the preprint is focused on describing the GECI expression conditions prone to generating these artefacts. Studying the effects of Ca2+ micro-waves on the circuitry are scientific questions, and would require an experimental framework of testing the aberrant activity on a specific physiological function e.g. place activity or specific oscillations (e.g. sharp-wave activity). Ca2+ microwaves, as the ones described here, have not been reported under physiological conditions or pathophysiological conditions and studying the effects of such artefactual waves on the circuit was not our intention.

With respect to place cell activity, specifically, it is intuitive that during the Ca2+ micro-wave the participating cell’s place field activity would be obscured by the artefactual activity. Cell activity appears to return immediately following the wave suggesting that the cells could exhibit place activity outside their participation in the Ca2+ micro-waves. However, we do not know if the Ca2+ micro-wave activity disrupts the generation or maintenance of place fields. We have now added a brief reference to possible effects on place coding to the paper (P12, L315-317).

The CaMKII-Cre approach for flexed-syn-GCaMP expression shows no micro-waves and is convincing, but it is only from 2 animals, even though both had no micro-waves.In light of the reviewer’s comment, we have added a further 3 animals with conditional expression of GCaMP6m from the DZNE to complement the current dataset with conditional expression of GCaMP6s from UoB (P10, L236 & 239 and revised table 1). Although Ca2+ waves were not observed in any of the in total 5 animals, we still do not know with all certainty whether this approach is completely safe. Time will show if researchers still encounter the phenotype under certain conditions when using this conditional approach.The authors state in their Discussion that even without observable microwaves, a syn-Ca2+-indicator transduction strategy could still be problematic. This may be true, but they do not check this in their analysis, so it remains unknown

We agree with the reviewer and have now made this point clearer in the revised discussion (P11, L257-258)

**Reviewer #3 (Public Review):**
Weaknesses:I believe that the weaknesses of the manuscript are appropriately highlighted by the authors themselves in the discussion. I would, however, like to emphasize several additional points.As the authors state, the exact conditions that lead to Ca2+ micro-waves are unclear from this manuscript. It is also unclear if Ca2+ micro-waves are specific to GECI expression or if high-titer viral transduction of other proteins such as genetically encoded voltage indicators, static fluorescent proteins, recombinases, etc could also cause Ca2+ micro-waves.

The high expression of other proteins has been shown to result in artefactual phenomenon such as toxicity or fluorescent puncta (for GFP see Hechler et al. 2006; Katayama et al. 2008 for GEVI see Rühl et al. 2021), but we are not aware of reports of micro-waves. Although it is certainly possible that high expression levels of other proteins could lead to waves, we suspect the Ca2+ micro-waves observed in this preprint result from a dysregulation of Ca2+ homeostasis. This is not to suggest that voltage indicators could not result in micro-waves (e.g. Ca2+ homeostasis may be indirectly affected).

The authors almost exclusively tested high titer (>5x10^12 vg/mL) large volume (500-1000 nL) injections using the synapsin promoter and AAV1 serotypes. It is possible that Ca2+ micro-waves are dramatically less frequent when titers are lowered further but still kept high enough to be useful for in vivo imaging (e.g. 1x10^12 vg/mL) or smaller injection volumes are used. It is also possible that Ca2+ micro-waves occur with high titer injections using other viral promoter sequences such as EF1α or CaMKIIα. There may additionally be effects of viral serotype on micro-wave occurrence.

We agree with all points raised by the reviewer. Notably, we used viral transduction protocols with titers and volumes within in the range of those previously used for viral transduction of GCaMP under the synapsin promoter (see P11 L269-275) and we observed Ca2+ micro-waves. As the reviewer suggested, we did find that lowering the titer is an important factor in reducing these Ca2+ micro-waves and there is likely a wide range of approaches that avoid the phenomenon. With regards to viral serotype, we show that micro-waves occurred across AAV1 and 9, but it is possible that other serotypes may avoid the phenomenon.

We reiterate in the abstract of the revised manuscript that expression level is a crucial factor (P2, L40 and P2, L44-45) and now mention that other promoters and induction protocols that result in high Ca2+ indicator expression may result in Ca2+ micro-waves P12, L291-294.

The number of animals in any particular condition are fairly low (Table 1) with the exception of V1 imaging and thy1-GCaMP6 imaging. This prohibits rigorous comparison of the frequency of pathological calcium activity across conditions.

We have now added 3 more animals with conditional GCaMP6 expression. In total, the study contains 34 animals with viral injection into the hippocampus from different laboratories and under different conditions resulting in multiple groups. As such we are cognizant of the resulting limitations for statistical evaluation.

However, in light of the reviewer’s comment, we have now employed a generalized linear model tested on all the data to examine the relationship between the Ca2+ micro-wave incidence and the different factors. The multivariate GLM did find a significant relationship between Ca2+ micro-wave incidence and both viral dilution and weeks post injection (see below and revised manuscript P8, L189-193).

For injections into CA1 in the hippocampus (n = 28), a GLM found no relationship between Ca2+ micro-waves and each of the individual variables x (Ca-wave ~ x) ; viral dilution: z score = 1.14, deviance above null = 1.31, p = 0.254; post injection weeks: : z score = 1.18, deviance above null = 1.44, p = 0.239; injection volume: : z score = -0.76, deviance above null = 0.59, p = 0.45; construct: : z score = 1.18, difference in deviance above null = 1.44, p = 0.239.

However, a multivariable logistic GLM relating dilution and post injection weeks (Ca-wave ~ dilution + p.i_wks) showed that together both variables were significantly related to Ca2+ micro-waves (Deviation above null = 7.5; Dilution: z score = 2.18, p < 0.05; p.i_wks : z score = 2.22, p < 0.05).

**Recommendations For The Authors:**

**Reviewer #1 (Recommendations For The Authors):**
Results are straightforward and convincing. While a couple of ways to reduce the aberrant microwaves of calcium responses were demonstrated, delving into the functions of interneurons is crucial for a more comprehensive understanding of cellular causality.

As mentioned in the public response, disentangling cellular mechanism from technical requirements will need a large and systematic study. To determine the contribution from interneurons, the use of specific interneuron promoters would be required, and viral titers systematically varied to result in similar cellular GCaMP expression levels as seen under the synapsin promoter condition.

**Reviewer #2 (Recommendations For The Authors):**
Do the authors think the cells are firing when they participate in a micro-wave, or do they think the calcium influx is due to something else? A discussion point on this would be good.

This is an excellent point raised by the reviewer. We do not know if the elevated cellular Ca2+ during the artifactual Ca2+ micro-wave reflects action potential firing or an increase of Ca2+ from intracellular stores. As already described in the text of the preprint, their optical spatiotemporal profile neither fits with known microseizure progression patterns, nor with spreading depolarization/depression. We have adopted the reviewer’s suggestion and added the following point to the discussion section in the revised preprint (P12, L308-315):

In a limited dataset, we attempted to detect the Ca2+ micro-waves by hippocampal LFP recordings (using a conventional insulated Tungsten wire, diameter ~110µm). We could not identify a specific signature, e.g. ictal activity or LFP depression, which may correspond to these Ca2+ micro-waves. The crucial shortcoming of this experiment of course is that with these LFP recordings, we could not simultaneous perform hippocampal 2-photon microscopy. Thus, it is uncertain if the Ca2+ micro-waves indeed occurred in proximity to our electrode.

The results seem to suggest that micro-waves may involve interneurons as their CaMKII-Cre strategy avoids waves - possibly due to a lack of expression of GECIs in interneurons. It would be great to hear the author's thoughts on this and add a brief discussion point.

As mentioned in public response to Reviewer 1, it is difficult to disentangle cellular mechanisms from technical requirements, and the exact requirements for the Ca2+ micro-waves to occur are still not fully clear. The absence of Ca2+ micro-waves in our CaMKII-Cre dataset may indeed reflect the requirement of interneurons. However, it could just as well be due to a sparse labelling of principle cells or simply reflect differences in the expression levels of GCaMP under the different promotors.

All in all, a more complete understanding of the requirements of such Ca2+ micro-waves will require a community effort. Therefore, it is important that each group check the safety profile of their GECI and report problems to the community.

We have added these points to the revised preprint (P12, L291 and P12, L298)

Plotting the incidence of micro-waves as a function of the age of mice would be a nice addition (the authors have the data).

There was no relationship of Ca2+ micro-wave occurrence or frequency with age over the range of 5-79 wks (see public response) and this has been added to the preprint (P14, L354)

**Reviewer #3 (Recommendations For The Authors):**
I appreciate the authors raising the awareness of this issue. I had personally observed micro-waves in my own data as well. In agreement with their findings, I found that the occurrence of micro-waves was dramatically lower when I reduced the viral titer. Anecdotally, I also observed voltage micro-waves when virally transducing genetically encoded voltage indicators at similar titers. For that reason, I am skeptical that this issue is exclusive to GECIs.

We find it interesting that the reviewer has also seen artefactual micro-waves following viral transduction of genetically encoded voltage indicators. Without seeing the voltage waves the referee is referring to or the conditions, it is of course difficult to compare with the Ca2+ micro-waves we report. However, this comment again raises the question of mechanism. We believe that in the GECI framework, Ca2+ homeostatic aspects are important. Voltage indicators are based on different sensor mechanisms, and expressed in the cell membrane, but it may very well be that there are overlapping factors between Ca2+ and voltage indicators that could trigger a similar, or even the same phenomenon in the end.

Minor comments:(1) Line 131-132: I believe the authors only tested for micro-waves in V1. This should be made clear in the results. It could be that micro-waves could occur in other parts of cortex with the same viral titers.

Both V1 and somatosensory cortex were tested as described in the methods (P15, L395-397), we have made this clearer in the revised preprint (P6, L138).

(2) There are no statistics associated with the data from Fig 1e.

We have now added statistics (P5, L126).

(3) The authors may be able to make a stronger claim about the pathological nature of the micro-waves if there are differences in the histology between the injected and non-injected hemispheres. For example, is there evidence of widespread cell death in the injected hemisphere (e.g. lower cell count, smaller hippocampal volume, caspase staining, etc).

We found no evidence of gross morphological changes to the hippocampus following viral transduction with no changes in CA1 pyramidal cell layer thickness or CA1 thickness (pyramidal cell layer thickness: 49 ± 12.5 µm ipsilateral and 50.3 ± 11.1 µm contralateral, n = 4, Student’s t-test p=0.89; CA1 thickness: 553.3 ± 14 µm ipsilateral and 555.8 ± 62 µm contralateral, n = 4, Student’s t-test p=0.94; 48 ± 13 weeks post injection at time of perfusion).

We have added this to the preprint (P5, L117-122)

(4) The broader micro-waves in the stratum oriens versus the stratum pyramidale are likely due to the spread of the basal dendrites of pyramidal cells. If the typical size of the basal dendritic arbor of CA1 pyramidal neurons is taken into account, does this explain the wider calcium waves in this layer.

Absolutely, great point, yes, we completely agree on this. It is likely the active neuropil (including dendritic arbour) are contributing to the apparent broader diameter. In addition, as evident in the video 5 cell somata in the stratum Oriens (possibly interneurons) are active and their processes also contribute.

We have now mentioned these points in the revised preprint (P5, L132)

(5) Lines 179-181: Is the difference in the prevalence of micro-waves between viral titers statistically significant?

Although we have a large number of animals in total (n = 34) with viral injection into the hippocampus, the number of animals in each condition, given the many factors, is low. We therefore used a generalized linear model to test the relationship between the Ca2+ micro-waves and the variables.

We have now added this analysis to the revised preprint (P8, L189-193)

(6) Lines 200-203: The CA3 micro-waves were only observed at one institution. The current wording is slightly misleading.

We agree and have changed this to be clearer (P9 L216)